# The modular mechanism of chromocenter formation in *Drosophila*

**Madhav Jagannathan[1][†]*, Ryan Cummings[2][†], Yukiko M Yamashita[2,3]***

[1]Life Sciences Institute, University of Michigan, Ann Arbor, United States; [2]Howard Hughes Medical Institute, University of Michigan, Ann Arbor, United States; [3]Department of Cell and Developmental Biology, University of Michigan, Ann Arbor, United States

**Abstract** A central principle underlying the ubiquity and abundance of pericentromeric satellite DNA repeats in eukaryotes has remained poorly understood. Previously we proposed that the interchromosomal clustering of satellite DNAs into nuclear structures known as chromocenters ensures encapsulation of all chromosomes into a single nucleus (Jagannathan et al., 2018). Chromocenter disruption led to micronuclei formation, resulting in cell death. Here we show that chromocenter formation is mediated by a 'modular' network, where associations between two sequence-specific satellite DNA-binding proteins, D1 and Prod, bound to their cognate satellite DNAs, bring the full complement of chromosomes into the chromocenter. *D1 prod* double mutants die during embryogenesis, exhibiting enhanced phenotypes associated with chromocenter disruption, revealing the universal importance of satellite DNAs and chromocenters. Taken together, we propose that associations between chromocenter modules, consisting of satellite DNA binding proteins and their cognate satellite DNA, package the *Drosophila* genome within a single nucleus.
DOI: https://doi.org/10.7554/eLife.43938.001

**\*For correspondence:**
madhavj@umich.edu (MJ);
yukikomy@umich.edu (YMY)

[†]These authors contributed equally to this work

## Introduction

Satellite DNAs are abundant tandem repeats that are ubiquitously found at centromeric and pericentromeric heterochromatin of eukaryotic chromosomes. Whereas it is well established that centromeric satellite DNA serves as a platform to form the kinetochore (*Sun et al., 1997*; *Sun et al., 2003*; *Willard, 1990*), the role of pericentromeric heterochromatin has been obscure despite its abundance that surpasses far beyond centromeric heterochromatin. Due to the lack of protein-coding ability and lack of conservation among species, satellite DNA has been repeatedly consigned to the status of genomic junk (*Ohno, 1972*; *Orgel and Crick, 1980*), even though they can constitute ~50% of eukaryotic genomes. Although satellite DNAs have been proposed to function in diverse cellular processes such as meiotic disjunction (*Dernburg et al., 1996*; *Hawley et al., 1992*), dosage compensation (*Menon et al., 2014*) and chromosome segregation (*Rošić et al., 2014*), these functions have often been restricted to specific satellite DNA repeats, cell types or organisms. Accordingly, a central principle underlying the ubiquity and abundance of satellite DNA in eukaryotes has remained poorly understood.

In a recent study using *Drosophila* and mouse cells as models, we have proposed a conserved function of satellite DNAs in maintaining the entire chromosomal complement in a single nucleus (*Jagannathan et al., 2018*). Our study indicated that pericentromeric satellite DNAs play a critical role in bundling multiple chromosomes, leading to the formation of 'chromocenters', cytological structures that have been recognized for ~100 years (*Figure 1A*) (*Jones, 1970*; *Jost et al., 2012*; *Pardue and Gall, 1970*). We have shown that *Drosophila melanogaster* D1 and the mouse HMGA1 bundle chromosomes by binding to their cognate satellite DNAs ({AATAT}$_n$ and major satellite,

respectively) and clustering them into chromocenters. Loss of chromocenters (i.e. defective bundling of chromosomes) due to mutation/depletion of these satellite DNA-binding proteins resulted in the formation of micronuclei, because unbundled chromosomes budded out of interphase nuclei. This was associated with extensive DNA damage, as has been observed with micronuclei in other systems

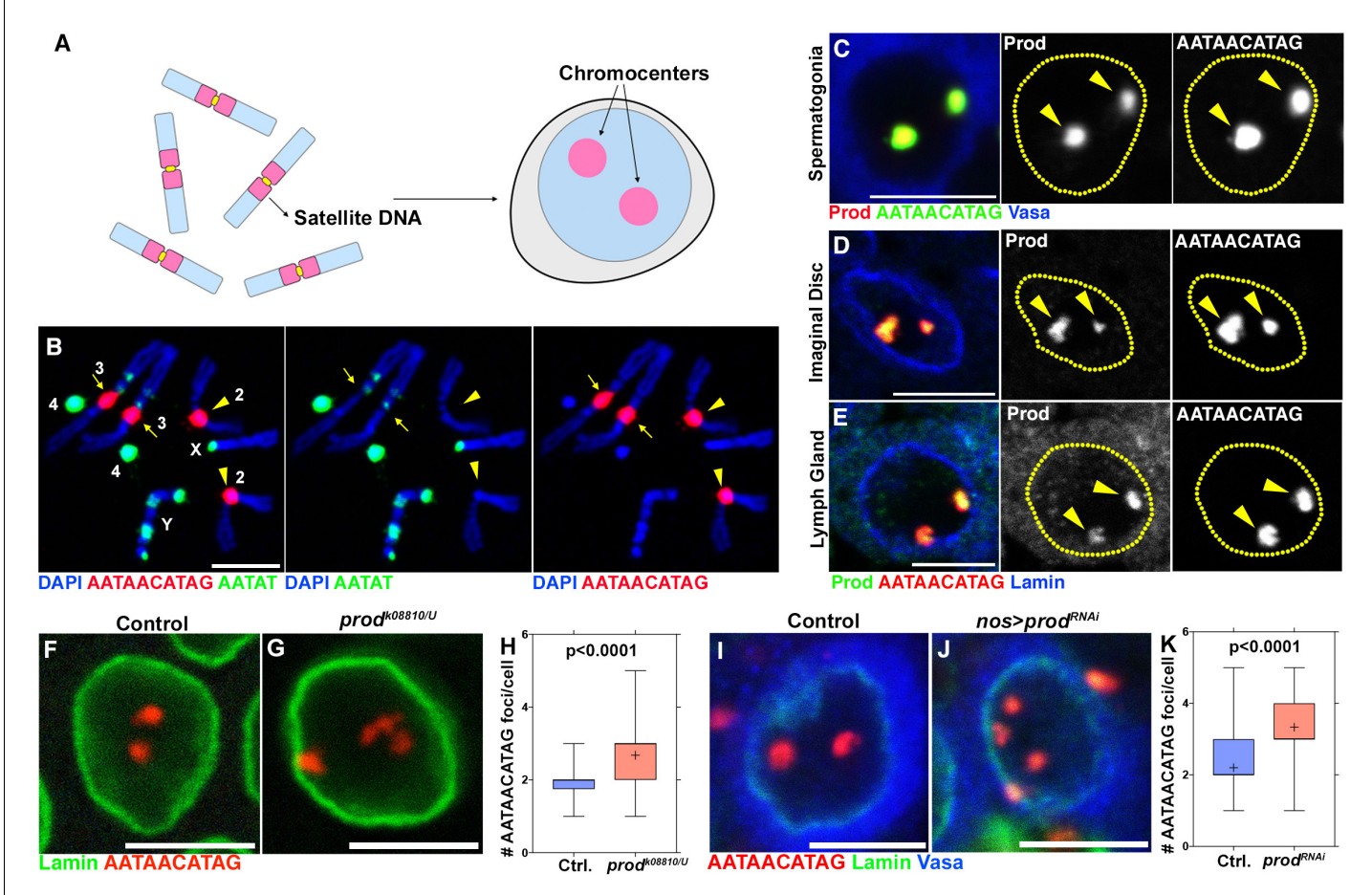

**Figure 1.** Prod bundles the {AATAACATAG}$_n$ satellite DNA on the *Drosophila melanogaster* major autosomes into chromocenters. (**A**) Schematic of chromosomes containing distinct pericentromeric satellite DNAs being organized into chromocenters. (**B**) FISH against the {AATAACATAG}$_n$ satellite (red) and the {AATAT}$_n$ satellite (green) on *Drosophila* larval neuroblast mitotic chromosomes co-stained with DAPI (blue) indicating the locations of these satellites in the *Drosophila* genome. Chromosome 2 (arrowheads) and chromosome 3 (arrows) were distinguished based on a small amount of {AATAT}$_n$ satellite on the chromosome 3. (**C**) FISH against the {AATAACATAG}$_n$ satellite (green) in a spermatogonium co-stained with Prod (red) and Vasa (blue). (**D, E**) FISH against the {AATAACATAG}$_n$ satellite (green) in larval imaginal disc cells (**D**) and larval lymph gland cells (**E**) co-stained with Prod (red) and Lamin (blue). (**F, G**) FISH against the {AATAACATAG}$_n$ satellite (red) in heterozygous control (**F**) and *prod*$^{k08810/U}$ (**G**) larval imaginal disc cells co-stained with Lamin (green). (**H**) Quantification of the number of {AATAACATAG}$_n$ chromocenters per larval imaginal disc cell (heterozygous control n = 70, *prod*$^{k08810/U}$ n = 71). (**I, J**) FISH against the {AATAACATAG}$_n$ satellite (red) in control (*nos-gal4/+; UAS-dcr-2/+*) (**I**) and Prod depleted (*nos-gal4/UAS-prod*$^{RNAi}$; *UAS-dcr-2/+*) (**J**) spermatogonia co-stained with Lamin (green) and Vasa (blue). (**K**) Quantification of {AATAACATAG}$_n$ chromocenters per spermatogonium (control n = 75, nos >*prod*$^{RNAi}$ n = 75). P values from Student's t-test are shown and crosshairs mark the mean. All scale bars are 5 μm.

DOI: https://doi.org/10.7554/eLife.43938.002

The following source data is available for figure 1:

**Source data 1.** Number of AATAACATAG foci/cell in control vs *prod* mutant imaginal discs (corresponding to *Figure 1H*).

DOI: https://doi.org/10.7554/eLife.43938.003

**Source data 2.** Number of AATAACATAG foci/cell in control vs *prod*$^{RNAi}$ spermatogonia (corresponding to *Figure 1K*).

DOI: https://doi.org/10.7554/eLife.43938.004

(*Crasta et al., 2012*; *Denais et al., 2016*; *Hatch et al., 2013*; *Raab et al., 2016*). Based on these observations, we proposed that chromocenter formation, supported by interchromosomal bundling, secures the full complement of chromosomes within a single nucleus.

Our previous study raised a few key questions. First, although some species such as mouse have the same pericentromeric satellite DNA repeat (i.e. major satellite) on all chromosomes, other species such as humans, *Drosophila*, cows and kangaroo rats are known to have divergent pericentromeric satellite DNA sequences on each chromosome (*John and Miklos, 1979*), raising the question as to how all chromosomes are bundled into chromocenters. For example, in *Drosophila melanogaster*, the $\{AATAT\}_n$ satellite is abundant on X, Y and $4^{th}$ chromosomes (*Figure 1B*), and we have shown that the D1 protein, which recognizes this satellite DNA, plays a critical role in chromocenter formation. However, chromosome 2 possesses little if any of the $\{AATAT\}_n$ satellite DNA (*Figure 1B*, arrowheads), and chromosome 3 possesses much less of this satellite DNA compared to X, Y and $4^{th}$ chromosomes (*Figure 1B*, arrows). Therefore, how chromosomes 2 and 3 (hereafter referred to as the major autosomes) may participate in chromocenter formation remained unclear. Second, it is widely observed that most eukaryotic cells contain multiple chromocenters, recognized as DAPI-dense foci or the association of pericentromeric satellite DNA. However, our model for chromocenter function necessitates the bundling of all chromosomes together, but how multiple chromocenter foci can mediate the bundling of all chromosomes remained unclear.

Here we show that the *Drosophila melanogaster* major autosomes are incorporated into chromocenters by the Proliferation disrupter (Prod) satellite DNA-binding protein and its cognate $\{AATAA-CATAG\}_n$ satellite DNA. Loss of Prod resulted in phenotypes similar to those of *D1* mutant with a different spectrum of affected tissues: defective chromocenters, micronuclei formation and loss of cellular viability in the imaginal discs and lymph glands. We show that D1 and Prod are mutually dependent in forming chromocenters, and our analysis implies that D1 and Prod might dynamically interact. These results suggest a network-like configuration for chromocenters, ensuring effective bundling of all *Drosophila* chromosomes. Moreover, a double mutant of *D1* and *prod* resulted in embryonic lethality with defective embryos exhibiting a striking increase in micronuclei, revealing the essential function of chromocenters and satellite DNAs for cell viability. Taken together, we propose that modules of satellite DNA and satellite DNA-binding proteins form chromocenters, which bundle the full complement of an organism's chromosomes in a single nucleus.

## Results

### The $\{AATAACATAG\}_n$ satellite DNA-binding protein, Prod, is important for chromocenter formation

Our previous study demonstrated that the *Drosophila* D1 protein, which binds to the $\{AATAT\}_n$ satellite DNA, plays a critical role in bundling chromosomes into chromocenters, such that they are encapsulated within a single nucleus. Puzzlingly, little if any $\{AATAT\}_n$ satellite DNA is present on $2^{nd}$ chromosomes (*Figure 1B*), raising a question as to how these autosomes may be incorporated into chromocenters.

The $\{AATAACATAG\}_n$ satellite DNA drew our attention, because it is abundantly present on chromosomes 2 and 3 (*Figure 1B*) and is known to be bound by the Proliferation disrupter (Prod) protein (*Török et al., 1997*; *Török et al., 2000*; *Platero et al., 1998*). Indeed, we observed that the $\{AATAACATAG\}_n$ satellite DNA and Prod protein perfectly co-localized within interphase nuclei in multiple cell types such as spermatogonia, imaginal disc cells and lymph gland cells (*Figure 1C–E*). Whereas the $\{AATAACATAG\}_n$ satellite DNA exists at four loci in diploid cells (2 from $2^{nd}$, 2 from $3^{rd}$) (*Figure 1B*), we predominantly observed two $\{AATAACATAG\}_n$ satellite DNA foci per nucleus (*Figure 1C–E*), suggesting that this satellite DNA is bundled into chromocenters. The number of $\{AATAACATAG\}_n$ satellite DNA foci per nucleus significantly increased in cells depleted of Prod (both loss-of-function mutants and RNAi) (*Figure 1F–K* and *Figure 2—figure supplement 1A–B*), suggesting that Prod functions to bundle $\{AATAACATAG\}_n$ satellite DNA into chromocenters.

## *prod* mutant cells exhibit cellular phenotypes associated with chromocenter disruption, leading to larval lethality

Loss-of-function of *prod* was reported to cause late larval lethality with these mutants containing atrophied imaginal discs and melanized lymph glands (*Török et al., 1997*). However, the cellular phenotypes in these degenerate tissues have not been investigated in detail. Because we observed defective chromocenter formation in *prod* mutant imaginal discs cells (*Figure 1F–H*), we wondered if larval lethality may be explained by cellular defects due to chromocenter disruption.

Our prior study showed that disruption of chromocenter formation in *D1* mutants led to micronuclei formation due to lack of chromosome bundling in interphase nuclei, leading to a loss of cellular viability in the *Drosophila* germline (*Jagannathan et al., 2018*). Similar to these observations with *D1* mutant germ cells, we observed that mutation of *prod* resulted in the formation of micronuclei in larval imaginal discs (*Figure 2A–B,E*) and lymph glands (*Figure 2C–E*). These micronuclei almost always contained satellite DNA (*Figure 2F*, >80% of micronuclei contained at least one of {AATAT}$_n$ or {AATAACATAG}$_n$ satellite DNAs (n = 48)), supporting the idea that micronuclei formation is due to declustering of chromocenters.

It is well established that the DNA within micronuclei is prone to genomic instability including excessive levels of DNA damage (*Crasta et al., 2012*; *Hatch et al., 2013*). We have shown that this is the case with *D1* mutant germ cells in *Drosophila*, leading to cell lethality (*Jagannathan et al., 2018*). We therefore quantified cells containing DNA damage assessed by anti-γ-H2Av antibody staining in *prod* mutant imaginal discs and lymph glands. We did not observe a significant difference in DNA damage levels between heterozygous control and *prod* mutant tissues (control imaginal disc cells – 0%, n = 161, *prod* mutant imaginal disc cells – 1.9%, n = 159, control lymph gland cells – 0.7%, n = 280 and *prod* mutant lymph gland cells – 1.4%, n = 143). However, when apoptotic cell death was blocked by expressing the death-associated inhibitor of apoptosis 1 (DIAP1) protein (*Orme and Meier, 2009*), we observed a striking increase in DNA damage in *prod* mutant tissues (*Figure 2G–J*), suggesting that *prod* mutation indeed increases DNA damage and that these damaged cells are rapidly cleared by apoptosis. Taken together, these results suggest that micronuclei formation in *prod* mutant tissues arising from chromocenter disruption leads to cell death due to elevated DNA damage, thereby resulting in the atrophy of essential somatic tissues and thus larval lethality.

We previously showed that *D1* mutation mainly affected germ cells (*Jagannathan et al., 2018*). To examine the role of *prod* in germ cells, we used RNAi-mediated knockdown (*Figure 2—figure supplement 1A–B*) and loss-of-function clones (*Figure 2—figure supplement 1C–D*). Germline depletion of *prod* resulted in disruption of {AATAACATAG}$_n$ satellite DNA clustering (*Figure 1I–K*) and defects in nuclear integrity: 27.9% of *prod*-depleted germ cells (n = 190) exhibited leakage of nls-GFP outside the major nucleus in comparison to 3.7% of control cells (n = 191) (*Figure 2—figure supplement 1E–F*). However, we did not observe a dramatic loss of germ cell viability (*Figure 2—figure supplement 1A–D,G*) or an increase in micronuclei formation in *prod*-depleted testes (no micronuclei were observed in both control and *prod*$^{RNAi}$ cells, n = 200 in both conditions). Conversely, *D1* mutation, which results in severe micronuclei formation and cell death in the germline, did not show a significant increase in micronuclei formation in somatic tissues that are affected by *prod* mutation such as imaginal discs and lymph glands (*Figure 2—figure supplement 2A–C*). These results show that while both *D1* and *prod* are important chromocenter-forming proteins, their depletion affects distinct tissues. Despite the distinct tissue requirement of these genes, *D1 prod* double mutant exhibited synthetic lethality (see below) suggesting that these genes function in the same biological process. At the moment, the underlying cause(s) of tissue specificity remain elusive although it is possible that tissue-specific nuclear organization may make each cell type more or less sensitive to the perturbation of X-Y-4 chromosome bundling vs major autosome bundling.

## Ectopic expression of Prod bundles heterologous chromosomes in spermatocytes

The above results show that Prod, like D1, plays a critical role in chromocenter formation. Based on our previous observation that D1 protein forms 'chromatin threads' that connect heterologous chromosomes (*Jagannathan et al., 2018*), we postulated that these chromatin threads are the basis for chromocenter formation. Similarly, we observed chromatin threads with Prod protein (*Figure 3A*,

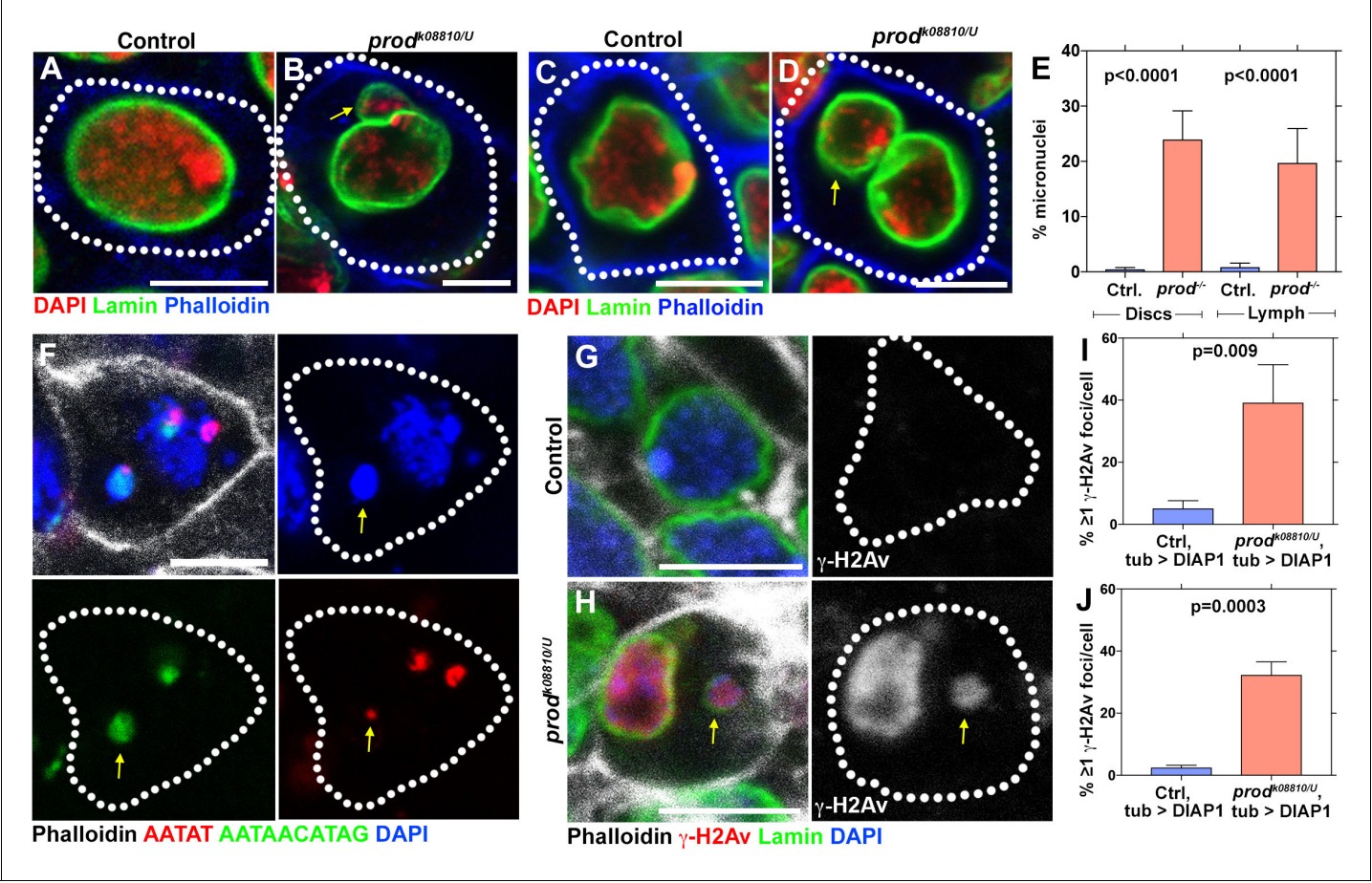

**Figure 2.** Loss of *prod* results in micronuclei formation and elevated DNA damage in larval imaginal discs and lymph glands. (A, B) Heterozygous control (A) and *prod*^{k08810/U} mutant (B) larval imaginal disc cells stained with DAPI (red), Lamin (green), and Phalloidin (blue). Arrow indicates micronucleus. (C, D) Heterozygous control (C) and *prod*^{k08810/U} mutant (D) larval lymph gland cells stained with DAPI (red), Lamin (green), and Phalloidin (blue). Arrow indicates micronucleus. (E) Quantification of micronuclei containing cells in heterozygous control and *prod*^{k08810/U} mutant imaginal discs and lymph glands. Control discs n = 744, *prod* mutant discs n = 501, Control lymph n = 664, *prod* mutant lymph n = 345. (F) FISH against the {AATAT}$_n$ satellite (red) and the {AATAACATAG}$_n$ satellite (green) in a *prod* mutant larval imaginal disc cell co-stained with Phalloidin (white) and DAPI (blue). Arrow indicates the presence of the third chromosome in the micronucleus. (G, H) Heterozygous control (G) and *prod*^{k08810/U} mutant (H) larval imaginal disc cells stained with Phalloidin (white), γ-H2Av (red), Lamin (green), and DAPI (blue). Arrow indicates the micronucleus containing DNA damage (γ-H2Av). Note that major nuclei also contain DNA damage upon chromocenter disruption as was observed in *D1* mutant (*Jagannathan et al., 2018*). (I, J) Quantification of cells containing ≥1 γ-H2Av foci in heterozygous control and *prod*^{k08810/U} imaginal discs (I) and lymph glands (J). In all cases (panels G, (H, I, J), apoptotic cell death was blocked by expressing *UAS-DIAP1* with a ubiquitous *tub-gal4* driver. Control discs n = 577, *prod* mutant discs n = 278, Control lymph n = 330, *prod* mutant lymph n = 216. P values from student's t-test are shown. All error bars: SD. All scale bars: 5 μm.
DOI: https://doi.org/10.7554/eLife.43938.005

The following source data and figure supplements are available for figure 2:

**Source data 1.** Frequency of micronuclei in control vs *prod* mutant imaginal disc and lymph gland cells (corresponding to *Figure 2E*).
DOI: https://doi.org/10.7554/eLife.43938.006

**Source data 2.** Frequency of cells containing γ-H2Av foci in control vs *prod* mutant imaginal discs expressing DIAP1 (corresponding to *Figure 2I*).
DOI: https://doi.org/10.7554/eLife.43938.007

**Source data 3.** Frequency of cells containing γ-H2Av foci in control vs *prod* mutant lymph glands expressing DIAP1 (corresponding to *Figure 2J*).
DOI: https://doi.org/10.7554/eLife.43938.008

**Figure supplement 1.** Depletion of *prod* in spermatogonial cells does not result in micronuclei or loss of cellular viability.
DOI: https://doi.org/10.7554/eLife.43938.009

**Figure supplement 1—source data 1.** GSC number/testis in control vs prod RNAi testes (corresponding to *Figure 2—figure supplement 1G*).
DOI: https://doi.org/10.7554/eLife.43938.010

**Figure supplement 2.** Loss of *D1* results in low levels of micronuclei formation in larval imaginal discs and lymph glands.
DOI: https://doi.org/10.7554/eLife.43938.011

*Figure 2 continued on next page*

*Figure 2 continued*

**Figure supplement 2—source data 1.** Frequency of cells containing micronuclei in control vs *D1* mutant cells in imaginal discs and lymph glands (corresponding to *Figure 2—figure supplement 2C*).

DOI: https://doi.org/10.7554/eLife.43938.012

arrowheads indicate two of the four Prod loci while the arrow indicates Prod threads connecting the two loci). However, these D1/Prod-positive chromatin threads are only apparent during a narrow time window during mitotic prophase when individual chromosomes start to resolve from each other during chromosome condensation but prior to metaphase. By metaphase, the chromatin threads

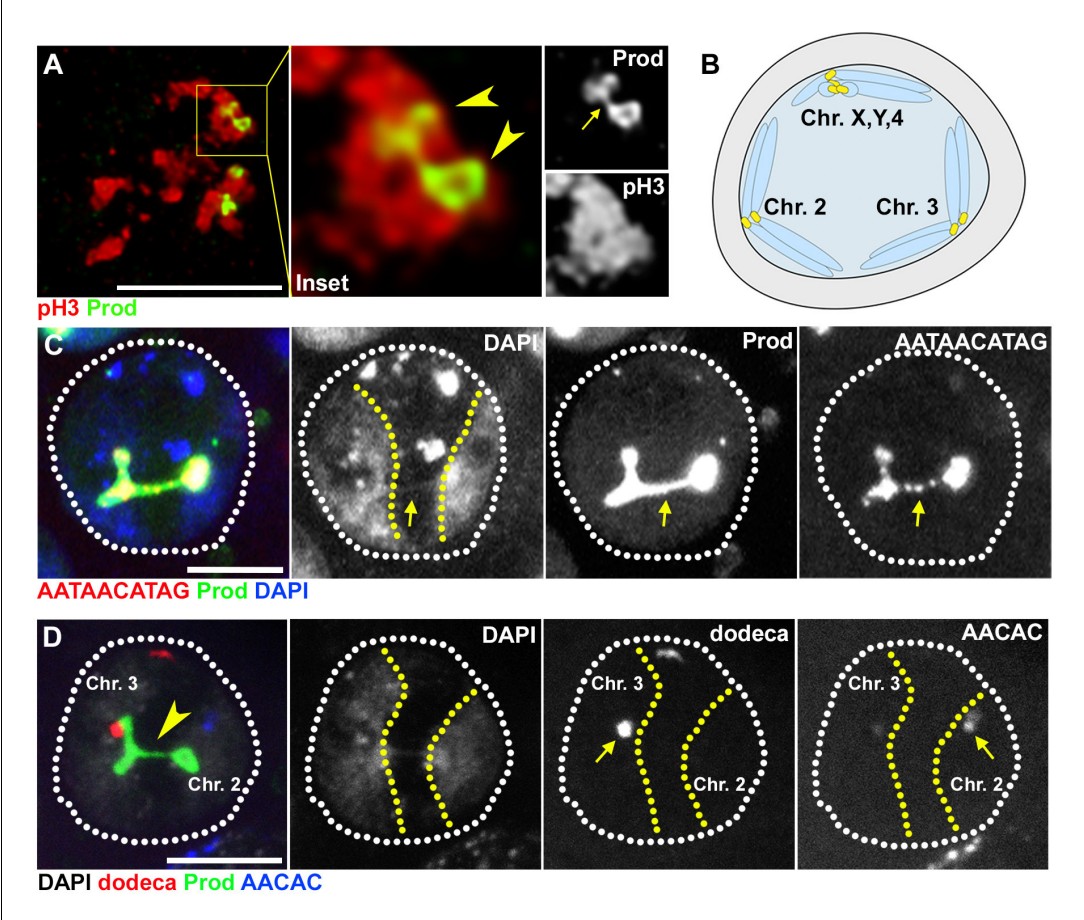

**Figure 3.** Prod bundles heterologous chromosomes through the {AATAACATAG}$_n$ satellite DNA. (**A**) Deconvolution microscopy of *Drosophila* larval neuroblasts in early prophase expressing GFP-Prod (green) under the control of *wor-gal4* and stained with pH3 (Ser10) (red). Arrowheads in the inset indicate two of the four Prod loci while the arrow indicates Prod threads connecting the two loci. (**B**) Schematic depicting how homologous chromosomes separate into distinct 'territories' in spermatocytes in preparation of meiotic reductional division. One territory contains the X, Y, and 4th chromosomes, a second territory contains the 2nd chromosomes, and a third territory contains the 3rd chromosomes. (**C**) FISH against the {AATAACATAG}$_n$ satellite (red) in spermatocytes expressing GFP-Prod (green) under the control of *bam-gal4* and stained with DAPI (blue). Yellow line demarcates chromosome 2 and chromosome 3 territories. Arrow indicates chromatin thread linking the two territories that is positive for DAPI, Prod, and the {AATAACATAG}$_n$ satellite DNA. (**D**) FISH against the dodeca satellite (red) and the {AACAC}$_n$ satellite (blue) in spermatocytes GFP-Prod (green) under the control of *bam-gal4* and stained with DAPI (white). Yellow line demarcates chromosome 2 and chromosome 3 territories. Arrowhead indicates chromatin thread linking the two territories that is positive for DAPI and Prod. Arrows indicate location of {AACAC}$_n$ satellite DNA that is specific to chromosome 2 and dodeca satellite DNA that is specific to chromosome 3. All scale bars are 5 µm.

DOI: https://doi.org/10.7554/eLife.43938.013

The following figure supplement is available for figure 3:

**Figure supplement 1.** Prod expression is not observed in *Drosophila* spermatocytes.

DOI: https://doi.org/10.7554/eLife.43938.014

that connect heterologous chromosomes were completely resolved, likely to allow mitotic chromosome segregation. Also it is not possible to visualize the clustering of heterologous chromosomes within chromocenters during interphase, as decondensed chromatin does not allow distinction of individual chromosomes.

Strikingly, we found that ectopic expression of GFP-Prod in spermatocytes, which normally lack Prod (*Figure 3—figure supplement 1*), led to the formation of chromatin threads between heterologous chromosomes. It is well established that homologous chromosomes separate into distinct 'territories' within the nucleus in preparation of meiotic reductional division in spermatocytes of *Drosophila* (*Figure 3B*) (*McKee, 2004*). Ectopic expression of GFP-tagged Prod protein in spermatocytes resulted in the formation of chromatin threads between distinct chromosome territories (*Figure 3C*) with these bridges positive for both Prod protein and the {AATAACATAG}$_n$ satellite DNA (*Figure 3C*, arrows indicate Prod-{AATAACATAG}$_n$ threads connecting chromosome territories marked by the yellow dashed lines). The two chromosomal territories connected by ectopically expressed Prod are territories containing chromosomes 2 and 3, as determined by FISH using $2^{nd}$ and $3^{rd}$ chromosome specific probes ({AACAC}$_n$ satellite for 2, dodeca satellite for 3) (*Jagannathan et al., 2017*) (*Figure 3D*). These data clearly demonstrate that ectopically expressed Prod bound the {AATAACATAG}$_n$ satellite DNA on chromosomes 2 and 3, and physically linked these chromosomes. We conclude that Prod is sufficient to bundle heterologous chromosomes via binding its cognate satellite DNA, {AATAACATAG}$_n$, providing a mechanistic explanation how chromocenters are formed by bundling of specific satellite DNA sequences located on heterologous chromosomes.

## D1 and Prod foci dynamically associate in interphase nuclei

The results described above and our previous study establish that D1 and Prod are required to cluster pericentromeric satellite DNAs to form chromocenters. D1 and Prod each bundle chromosomes that contain their cognate satellite DNA. However, this raises the question as to how the entire set of chromosomes can be bundled together such that they are encapsulated in the same nucleus. We noted that D1 and Prod were predicted to interact with each other, albeit weakly, based on a high-throughput mass spectrometry screen in Schneider S2 cells (*Guruharsha et al., 2011*), suggesting that satellite DNA bundled by D1 and that bundled by Prod might further cluster together via an interaction between Prod and D1 proteins. However, co-immunoprecipitation experiments did not detect any physical interaction between these proteins (*Figure 4—figure supplement 1*), suggesting that their interaction may be weak or transient. We observed that D1- and Prod-positive foci were consistently juxtaposed within the heterochromatic domain of the nucleus (*Figure 4A*, the heterochromatic domain is demarcated by the yellow dashed line and is based on HP1 localization). This pattern of D1 and Prod localization was observed in multiple cell types (*Figure 4—figure supplement 2A–C*, arrows indicate juxtaposed foci). Based on the weak interaction detected by immunoprecipitation/mass-spectrometry and juxtaposition of D1 and Prod foci within the nucleus, we hypothesized that D1 and Prod may interact transiently.

To examine this possibility, we conducted time-lapse live observation by combining GFP-D1 and mCherry-Prod expressed in spermatogonial cells. While we have observed an average of ~4–5 D1-positive foci/nucleus and ~2 Prod positive foci/nucleus in fixed and stained samples, the live observation demonstrated that these foci are not static but rather moving dynamically within the nucleus (*Figure 4B–C*). Using single particle tracking, we estimated the diffusion coefficient (D), which indicated that D1 and Prod exhibited similar dynamics (*Figure 4D*). We also quantified the slope of momentum scaling spectrum (S$_{MSS}$) of D1 and Prod foci, an established parameter for the type of particle movement (*Ewers et al., 2005*). In the case of free, unhindered diffusion, the S$_{MSS}$ value equals 0.5 whereas values below and above 0.5 indicate confined and directed motion respectively. The majority of D1 and Prod foci exhibited S$_{MSS}$ <0.5 (*Figure 4E*), suggesting confined motion, likely within the heterochromatin domain. D1-positive foci and Prod-positive foci frequently became juxtaposed in a 'kiss and run' manner: coming into contact temporarily and separating from each other soon after (*Figure 4F–G*). These results indicate that chromocenter is a structure comprised of dynamic modules of satellite DNAs and satellite DNA binding proteins, rather than rigidly linked satellite DNAs. The dynamism exhibited by these satellite DNA binding proteins may reflect the recently demonstrated liquid droplet-like properties of heterochromatin (*Larson et al., 2017*; *Strom et al., 2017*).

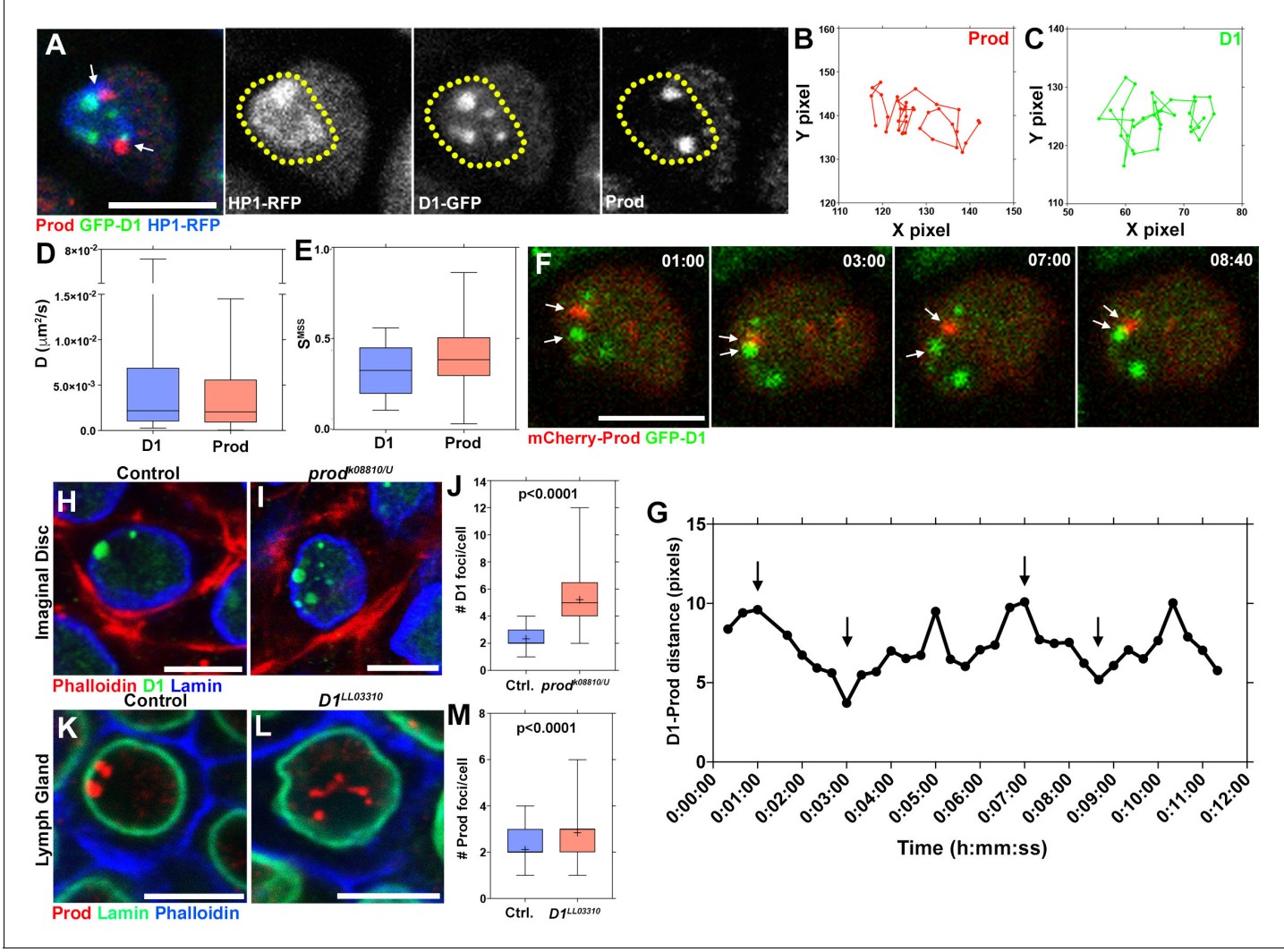

**Figure 4.** Dynamic association between D1 and Prod mediates the formation of chromocenters. (A) *Drosophila* lymph cells expressing D1-GFP (green) and HP1-RFP (blue) stained with Prod (red). Arrows indicate juxtaposed Prod and D1 foci. Yellow line demarcates the heterochromatic domain based on HP1 localization. Scale bar: 5 µm. (B, C) Particle tracking analysis of single Prod (B) or D1 (C) foci in the XY plane. (D, E) Box-and-whisker plot of the diffusion co-efficients (D) and the slope of momentum scaling spectrum (E) of D1 (n = 25) and Prod (n = 29). (F) Time-lapse imaging of spermatogonia expressing *nos-gal4* driven UAS-mCherry-Prod (red) and D1-GFP (green). Arrows indicate Prod and D1 foci coming together in a 'kiss-and-run' manner. Time is indicated in mm:ss. (G) Quantification of the distance between the D1 and Prod foci indicated in panel F over time. Arrows indicate the time points shown in panel F. (H, I) Heterozygous control (H) and *prod^{k08810/U}* mutant (I) larval imaginal disc cells stained with Phalloidin (red), D1 (green), and Lamin (blue). (J) Box-and-whisker plot of the number of D1 foci per larval imaginal disc cell (control n = 72, *prod^{k08810/U}* n = 65). (K, L) Heterozygous control (K) and *D1^{LL03310}* mutant (L) larval lymph gland cells stained with Prod (red), Lamin (green), and Phalloidin (blue). (M) Box-and-whisker plot of the number of Prod foci per larval lymph gland cell (control n = 63, *D1^{LL03310}* n = 66). All P values are from student's t-test. All middle bars: median. All crosshairs: mean. All scale bars: 5 µm.

DOI: https://doi.org/10.7554/eLife.43938.015

The following source data and figure supplements are available for figure 4:

**Source data 1.** Numerical data of particle tracking for Prod foci (corresponding to *Figure 4B*).
DOI: https://doi.org/10.7554/eLife.43938.016
**Source data 2.** Numerical data of particle tracking for D1 foci (corresponding to *Figure 4C*).
DOI: https://doi.org/10.7554/eLife.43938.017
**Source data 3.** Diffusion co-efficients of D1 and Prod (corresponding to *Figure 4D*).
DOI: https://doi.org/10.7554/eLife.43938.018
**Source data 4.** Slope of momentum scaling spectrum of D1 and Prod (corresponding to *Figure 4E*).
DOI: https://doi.org/10.7554/eLife.43938.019
**Source data 5.** Measurements of D1-Prod distance (corresponding to *Figure 4G*).

*Figure 4 continued*

DOI: https://doi.org/10.7554/eLife.43938.020
**Source data 6.** Number of D1 foci/cell in control vs *prod* mutant imaginal discs (corresponding to *Figure 4J*).
DOI: https://doi.org/10.7554/eLife.43938.021
**Source data 7.** Number of Prod foci/cell in control vs *D1* mutant lymph glands (corresponding to *Figure 4M*).
DOI: https://doi.org/10.7554/eLife.43938.022
**Figure supplement 1.** Co-immunoprecipitation experiments from multiple tissue lysates did not detect an interaction between Prod and D1.
DOI: https://doi.org/10.7554/eLife.43938.023
**Figure supplement 2.** Mutually dependent clustering of D1 and Prod in multiple cell types.
DOI: https://doi.org/10.7554/eLife.43938.024
**Figure supplement 2—source data 1.** Number of D1 foci/cell in control vs *prod* mutant neuroblasts (corresponding to *Figure 4—figure supplement 2F*).
DOI: https://doi.org/10.7554/eLife.43938.025
**Figure supplement 2—source data 2.** Number of D1 foci/cell in control vs prod RNAi spermatogonia (corresponding to *Figure 4—figure supplement 2I*).
DOI: https://doi.org/10.7554/eLife.43938.026
**Figure supplement 2—source data 3.** Number of Prod foci/cell in control vs D1 mutant neuroblasts (corresponding to *Figure 4—figure supplement 2L*).
DOI: https://doi.org/10.7554/eLife.43938.027
**Figure supplement 2—source data 4.** Number of Prod foci/cell in control vs D1 mutant spermatogonia (corresponding *Figure 4—figure supplement 2O*).
DOI: https://doi.org/10.7554/eLife.43938.028
**Figure supplement 3.** Mutually dependent clustering of the {AATAT}$_n$ and {AATAACATAG}$_n$ in multiple cell types.
DOI: https://doi.org/10.7554/eLife.43938.029
**Figure supplement 3—source data 1.** Number of AATAACATAG foci/cell in control vs *D1* mutant imaginal discs (corresponding to *Figure 4—figure supplement 3G*).
DOI: https://doi.org/10.7554/eLife.43938.030
**Figure supplement 3—source data 2.** Number of AATAACATAG foci/cell in control vs *D1* mutant lymph gland (corresponding to *Figure 4—figure supplement 3H*).
DOI: https://doi.org/10.7554/eLife.43938.031

The frequent associations of D1 and Prod led us to examine whether these modules may exhibit inter-dependency in chromocenter formation. Indeed, we found that mutation of *prod* resulted in defective clustering of D1 protein within the nuclei of imaginal discs, neuroblasts and spermatogonial germ cells (*Figure 4H–J* and *Figure 4—figure supplement 2D–I*). Similarly, mutation of *D1* resulted in the disruption of Prod protein clustering in multiple tissues (*Figure 4K–M* and *Figure 4—figure supplement 2J–O*). FISH experiments confirmed the declustering of {AATAT}$_n$ in *prod* mutant, and that of {AATAACATAG}$_n$ in *D1* mutant cells (*Figure 4—figure supplement 3A–H*). Taken together, these results support the idea that modules of Prod, D1 and their cognate satellite DNA function in a mutually dependent manner to form chromocenters. We propose that this inter-dependency shapes a network-like configuration for chromocenters and forms the basis for bundling the 'entire set' of chromosomes, instead of bundling individual satellite DNAs.

## *D1 prod* double mutant exhibits embryonic lethality

While our data suggest that both D1 and Prod play a role in chromocenter formation, loss of either protein only resulted in tissue-specific cellular defects. To test the possibility that D1 and Prod might have partly redundant functions, we examined the development of *D1 prod* double mutants using loss-of-function alleles (*prod*$^{k08810}$/CyO, Act-GFP; *D1*$^{LL03310}$/TM3, Act-GFP). As wild type alleles of *prod* and *D1* were carried on the balancer chromosomes marked by GFP expression, only the double mutant animals are GFP-negative, whereas any animal that carries at least one wild-type allele of *D1* or *prod* are GFP-positive. We observed that *D1* and *prod* double mutants (i.e. GFP-negative animals) were underrepresented (far below expected frequency at 6.25%) in larval population in contrast to GFP-negative larvae from control parents (+/CyO, Act-GFP; +/TM3, Act-GFP) (*Figure 5A*). These data suggested that double mutant animals largely failed to develop past the embryo stage.

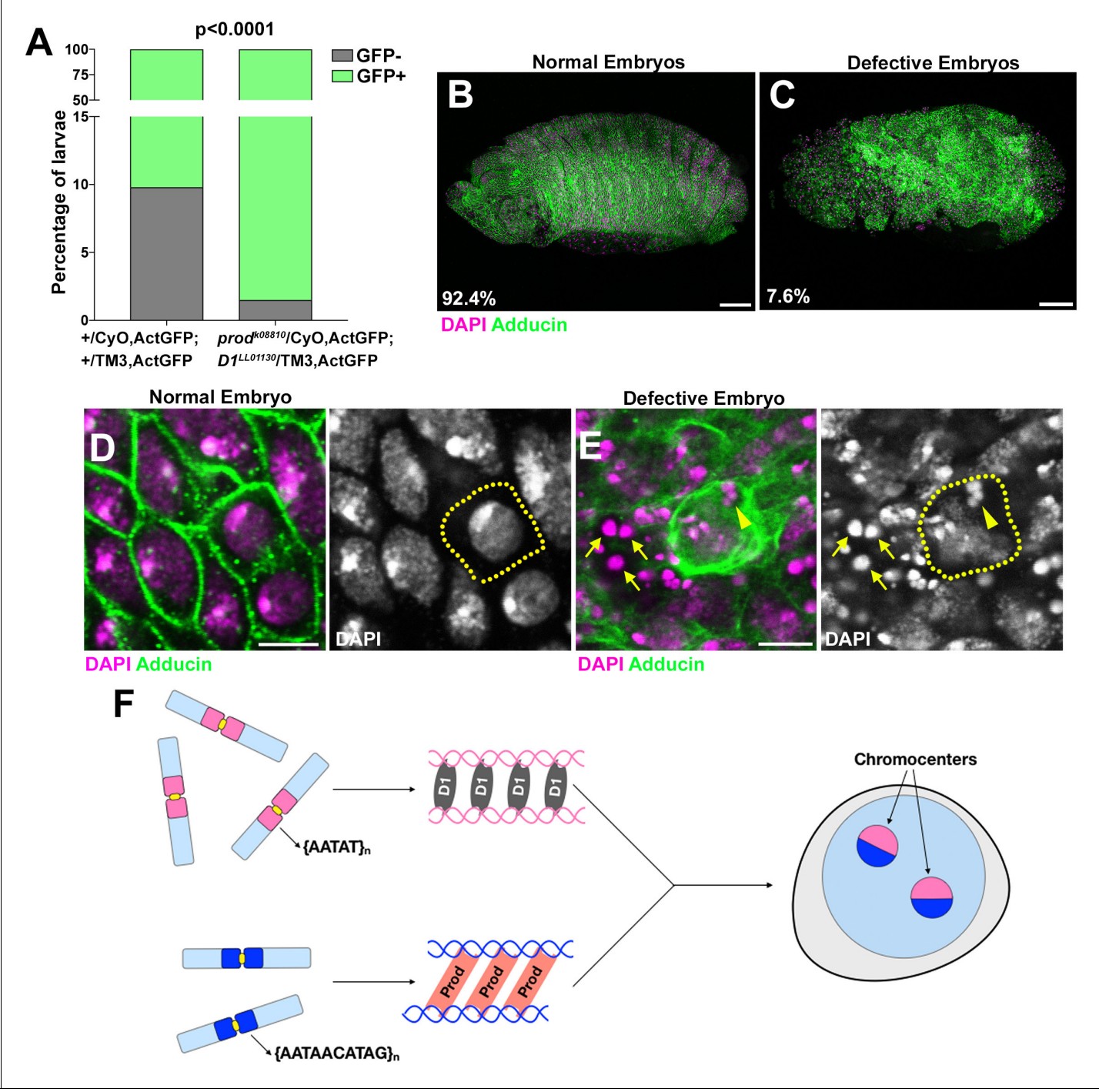

**Figure 5.** *D1 prod* double mutation leads to embryonic lethality. (**A**) Quantification of GFP-positive and GFP-negative first instar larvae sired by +/CyO, ActGFP; +/TM3, ActGFP and *prod*[k08810]/CyO, ActGFP; *D1*[LL03310]/TM3, ActGFP adults. GFP-negative larvae from +/CyO, ActGFP; +/TM3, ActGFP parents are wild type while GFP-negative larvae from *prod*[k08810]/CyO, ActGFP; *D1*[LL03310]/TM3, ActGFP parents are double mutants. P value from student's t-test is shown. (**B, C**) Normal (**B**) and defective (**C**) embryos from *prod*[k08810]/CyO, ActGFP; *D1*[LL03310]/TM3, ActGFP are stained with DAPI (red) and Adducin (green). Percentage of normal and defective embryos are indicated from n = 132. Scale bars: 25 μm. (**D, E**) Close-up view of normal (**D**) and defective (**E**) embryos from (**B, C**). Yellow line indicates cell boundary based on adducin staining. Arrows indicate extra-nuclear DNA and the arrowhead indicates micronuclei. Scale bar: 5 μm. (**F**) A model depicting the modular architecture of chromocenters in Drosophila *melanogaster*. Modules of D1-{AATAT}$_n$ and Prod-{AATAACATAG}$_n$ associate dynamically to bundle the entire chromosome complement into chromocenters.
DOI: https://doi.org/10.7554/eLife.43938.032

The following source data is available for figure 5:

*Figure 5 continued on next page*

*Figure 5 continued*

**Source data 1.** Percentages of GFP + vs GFP- larvae in the indicated genetic crosses (corresponding to *Figure 5A*).
DOI: https://doi.org/10.7554/eLife.43938.033

A closer examination revealed that 7.6% of embryos from mated *D1* and *prod* heterozygote parents exhibited abnormal development, (*Figure 5C*). This number closely matches with the expected frequency of 6.25% for double mutant embryos. We observed micronuclei in these defective embryos (*Figure 5D–E*) as well as numerous extranuclear DAPI-intense foci (*Figure 5E*, arrows), which we suggest is the terminal phenotype of cells with micronuclei due to chromocenter disruption. Taken together, our data establish the critical requirement of chromocenters and satellite DNA in maintaining the entire genome in a single nucleus and supporting cellular viability.

## Discussion

Satellite DNAs are one of the most abundant and ubiquitous elements of eukaryotic genomes. Nevertheless, these tandem repeats have often been dismissed as 'junk' DNA for the following reasons, (a) they are simple sequences with no apparent protein coding potential, (b) there is a striking lack of conservation in the abundance and identity of satellite DNA repeats even amongst closely related species. Nonetheless, satellite DNA abundance is remarkably stable over multiple generations, despite being prone to copy number loss through mechanisms such as replicative slippage and intra-chromatid exchange (*Charlesworth et al., 1994*), implying that satellite DNA must serve unappreciated function(s).

We have recently proposed a conserved function for pericentromeric satellite DNAs in encapsulating the entire chromosomal complement within a single nucleus (*Jagannathan et al., 2018*). The framework of this model is that satellite binding proteins, *Drosophila melanogaster* D1 and mouse HMGA1, bind their cognate satellite DNAs and bundle them into chromocenters. This bundling prevents individual chromosomes from budding off out of the nucleus, thereby maintaining chromosomes within the nucleus. Based on this idea, all the chromosomes must be bundled into chromocenters so as not to be lost as micronuclei. This raised a few questions. 1) How can all of the *D. melanogaster* chromosomes be bundled into chromocenters, given that the D1-bound $\{AATAT\}_n$ satellite DNA is present abundantly only on X, Y and $4^{th}$ chromosomes? and 2) If bundling of all chromosomes into chromocenters is required to package the genome into a single nucleus and given the multiple chromocenter foci typically observed in *Drosophila* and mouse cells, are these chromocenters linked to one another? In this study, through the investigation of Prod protein in *Drosophila melanogaster*, we provide insights into these questions.

First, we found that Prod, which is known to bind the $\{AATAACATAG\}_n$ satellite DNA on the major autosomes, is a chromocenter-forming protein, functioning together with D1 protein. Our study suggests that the chromocenter consists of multiple modules of satellite DNAs and satellite DNA-binding proteins, where D1 bundles X, Y and $4^{th}$ chromosomes and Prod bundles $2^{nd}$ and $3^{rd}$ chromosomes, thereby covering the full complement of the chromosomes (*Figure 5F*). Given that the *Drosophila melanogaster* genome contains at least 17 satellite DNAs (*Jagannathan et al., 2017*; *Lohe et al., 1993*), it is plausible that additional satellite DNA binding proteins may participate in chromocenter formation, even though D1 and Prod can cover the full complement of *D. melanogaster* chromosomes. Alternatively, these 17 satellite DNAs might reflect the history of the species, wherein individual satellites were essential at some point, while only a subset are critically important for the present day *D. melanogaster* genome.

We observed that D1-positive foci and Prod-positive foci dynamically associated within the interphase nuclei. D1 and Prod function interdependently such that D1/$\{AATAT\}_n$ clustering is dependent on Prod while Prod/$\{AATAACTAG\}_n$ clustering is dependent on D1. Therefore, although our data do not provide direct evidence that D1 and Prod physically interact to form chromocenters, we speculate the functional interaction of D1 and Prod plays a critical role in chromocenter formation. We further suggest that the association of satellite DNA binding proteins can form a chromocenter network, thereby packaging the entire genome within a single nucleus. Importantly, while mutation of either *D1* or *prod* exhibited varying effects on micronuclei formation in a tissue-specific manner, a

*D1 prod* double mutant resulted in a striking enhancement of the phenotype, leading to embryonic lethality. This suggests that while cells can compensate for the loss of an individual chromocenter module, loss of multiple chromocenter modules will result in more widespread and penetrant cellular defects, illuminating an essential function for chromocenter formation.

Our data suggest that the essence of satellite DNA function is to be bound by sequence-specific binding proteins that have the ability to bundle these repeats into chromocenters. As such, the satellite DNA sequence itself does not need to be conserved. This may explain why satellite DNA repeats diverge rapidly, even among closely related species, an observation that has led to the idea that satellite DNA is junk. For instance, the {AATAACATAG}$_n$ satellite DNA, which is a central player in bundling the *Drosophila melanogaster* major autosomes, is completely absent in its nearest relative, *Drosophila simulans*. These observations lead to an interesting speculation: if satellite DNA repeats diverge rapidly, their binding proteins will also likely adapt to attain optimal binding specificity for the diverged sequences. Whereas *D. melanogaster* Prod binds the {AATAACATAG}$_n$ satellite DNA, *D. simulans* Prod must have adapted to bind a distinct satellite DNA sequence to form chromocenters. This leads to a question as to whether *D. melanogaster* Prod can bundle *D. simulans* chromosomes and vice versa. It is tempting to speculate that such divergence in satellite DNA sequences and their binding proteins may lead to incompatibility in chromocenter formation when chromosomes from these two species are brought together in hybrids.

In summary, we propose that satellite DNA and their binding proteins conform to a modular system, whereby all the chromosomes are brought into a chromocenter network by the association of satellite DNA binding proteins. In this manner, chromocenter plays a fundamental role in securing all the chromosomes within a single nucleus.

## Materials and methods

### Fly husbandry and strains

All fly stocks were raised on standard Bloomington medium at 25°C. The following fly stocks were used: *prod^U* (BDSC42686), *UAS-GFP-nls* (BDSC4776), *tub-gal4* (BDSC5128), *hs-flp* (BDSC6), *FRT42D, Ubi-nls-GFP* (BDSC5626), *UAS-dcr-2* (BDSC24650), *D1-GFP* (BDSC50850), *HP1-RFP* (BDSC30562) and *UAS-DIAP1* (BDSC6657) were obtained from the Bloomington *Drosophila* stock center. *D1^LL03310* (DGRC140754) and *FRT42D prod^k08810* (DGRC111248) were obtained from the Kyoto stock center. *UAS-prod^RNAi* (VDRCv106593) was obtained from the Vienna Drosophila stock center. *nos-gal4* and *bam-gal4* have been previously described (*Chen and McKearin, 2003*; *Van Doren et al., 1998*). *wor-gal4* was a kind gift from Cheng-Yu Lee. Prod-null clones (indicated by loss of GFP signal) were generated as follows – Testes from flies of the genotype *hs-flp; FRT42D, Ubi-nls-GFP/ FRT42D, prod^k08810* were dissected 48 hr following a 1 hr heat shock at 37°C. For embryo and larval development analysis, flies laid eggs on apple-agar at RT and development was assessed every 24 hr.

### Transgene construction

For construction of *pUASt-GFP-Prod*, the *Prod* ORF was PCR-amplified from cDNA using the following primer pair, 5'-GTAGCGGCCGCAATGAACGGCAAGATG-3' and 5'-GTAGGTACCCTATAAG-GACGGCGGATCG-3'. The amplified fragment was subcloned into the NotI and KpnI sites of *pUASt-EGFP-attB* (*Salzmann et al., 2013*) resulting in *pUASt-GFP-Prod*. For construction of *pUASt-mCherry-Prod*, the mCherry ORF was PCR amplified from a plasmid template using the following primer pair, 5'-GTAGAATTCCATCGCCACCATGGTGAGCAAGGGCGAGGAG-3' and 5'-G TAGCGGCCGCCTTGTACAGCTCGTCCATGCC-3'. The amplified fragment was subcloned into the EcoRI and NotI sites of *pUASt-GFP-Prod*, replacing the GFP fluorophore, and resulting in *pUASt-mCherry-Prod*. Transgenic flies were generated by PhiC31 integrase-mediated transgenesis into the *attP2* site (BestGene).

### Immunofluorescence staining and microscopy

For *Drosophila* tissues, immunofluorescence staining was performed as described previously (*Cheng et al., 2008*). Briefly, tissues were dissected in PBS, transferred to 4% formaldehyde in PBS and fixed for 30 min. Tissues were then washed in PBS-T (PBS containing 0.1% Triton-X) for at least

60 min, followed by incubation with primary antibody in 3% bovine serum albumin (BSA) in PBS-T at 4°C overnight. Samples were washed for 60 min (three 20 min washes) in PBS-T, incubated with secondary antibody in 3% BSA in PBS-T at 4°C overnight, washed as above, and mounted in VECTA-SHIELD with DAPI (Vector Labs). For *Drosophila* embryos, 0–16 hr embryos were collected, dechorionated in 50% bleach, fixed and then devitellinized in methanol. The following primary antibodies were used: rabbit anti-vasa (1:200; d-26; Santa Cruz Biotechnology), mouse anti-LaminDm$_0$ (ADL84.12, 1:200, Developmental Studies Hybridoma Bank), mouse anti-$\gamma$−H2Av (UNC93-5.2.1, 1:400, Developmental Studies Hybridoma Bank), Phalloidin-Alexa546 (ThermoFisher, a22283, 1:200), rabbit anti-Prod (gift from Tibor Torok, 1:5000) and guinea pig anti-D1 (generated using the synthetic peptide CDGENDANDGYVSDNYNDSESVAA (Covance)). Images were taken using a Leica TCS SP8 confocal microscope with 63x oil-immersion objectives (NA = 1.4). Deconvolution was performed when indicated using the Hyvolution package from Leica. Images were processed using Adobe Photoshop software.

## Time-lapse live imaging

Testes from newly eclosed flies were dissected into Schneider's *Drosophila* medium containing 10% fetal bovine serum. The testis tips were placed inside a sterile glass-bottom chamber and were mounted on a three-axis computer-controlled piezoelectric stage. An inverted Leica TCS SP8 confocal microscope with a 63 $\times$ oil immersion objective (NA = 1.4) was used for imaging. Single particle tracking was performed using the MOSAIC suite plugin for ImageJ on 0.75 mm z-sections. We followed a previously established approach (*Ewers et al., 2005*; *Siebrasse et al., 2016*) to measure the dynamics of D1 and Prod foci. Both the diffusion co-efficient (D) and slope of the momentum scaling spectrum ($S_{MSS}$) were determined using the MOSAIC suites plugin. The MSS will show a straight line through the origin and its slope ($S_{MSS}$) is an excellent measure for the type of movement. In case of free, unconstrained diffusion, the slope is 0.5 while values above and below 0.5 indicate directed motion or confined motion respectively. All images were processed using Adobe Photoshop software.

## DNA fluorescence in situ hybridization

Whole mount *Drosophila* tissues were prepared as described above, and optional immunofluorescence staining protocol was carried out first. Subsequently, samples were post-fixed with 4% formaldehyde for 10 min and washed in PBS-T for 30 min. Fixed samples were incubated with 2 mg/ml RNase A solution at 37°C for 10 min, then washed with PBS-T +1 mM EDTA. Samples were washed in 2xSSC-T (2xSSC containing 0.1% Tween-20) with increasing formamide concentrations (20%, 40% and 50%) for 15 min each followed by a final 30 min wash in 50% formamide. Hybridization buffer (50% formamide, 10% dextran sulfate, 2x SSC, 1 mM EDTA, 1 µM probe) was added to washed samples. Samples were denatured at 91°C for 2 min, then incubated overnight at 37°C. For mitotic chromosome spreads, larval 3$^{rd}$ instar brains were squashed according to previously described methods (*Larracuente and Ferree, 2015*). Briefly, tissue was dissected into 0.5% sodium citrate for 5–10 min and fixed in 45% acetic acid/2.2% formaldehyde for 4–5 min. Fixed tissues were firmly squashed with a cover slip and slides were submerged in liquid nitrogen until bubbling ceased. Coverslips were then removed with a razor blade and slides were dehydrated in 100% ethanol for at least 5 min. After drying, hybridization mix (50% formamide, 2x SSC, 10% dextran sulfate, 100 ng of each probe) was applied directly to the slide, samples were heat denatured at 95°C for 5 min and allowed to hybridize overnight at room temperature. Following hybridization, slides were washed 3 times for 15 min in 0.2X SSC and mounted with VECTASHIELD with DAPI (Vector Labs). The following probes were used for *Drosophila* in situ hybridization: {AATAT}$_6$, {AACAC}$_6$, {dodeca}, {AATAACATAG}$_3$ and have been previously described (*Jagannathan et al., 2017*).

## Immunoprecipitation

*Drosophila* testis lysate was obtained from Upd tumor testes expressing either GFP or GFP-Prod under the control of *nos-gal4* while *Drosophila* brain lysate was obtained from third instar larval brains expressing GFP or GFP-Prod under the control of *wor-gal4*. Both sets of tissues were dissected into Schneider's *Drosophila* Medium (Thermo Fisher, 21720–024). Tissue were lysed in a buffer containing 50 mM Tris (pH 7.5), 5% glycerol, 0.2% IGEPAL, 1.5 mM MgCl$_2$, 125 mM NaCl

supplemented with 1 mM PMSF and protease inhibitor cocktail (Sigma, P8340) on ice for 30 min. Lysates were then centrifuged at 14,000 rpm for 15 min at 4°C and the supernatant was incubated with GFP-Trap beads (ChromoTek, gtmak-20) for 3 hr at 4°C by rotating the tubes end-over-end. The beads were washed three times with the supplied wash buffer and boiled for 10 min at 95°C in 2x SDS-sample buffer to dissociate immunoprecipitated proteins from the beads.

### SDS-PAGE and western blotting

SDS-PAGE and Western blotting were used to analyze the immunoprecipitated proteins. Samples were run on 10% Tris-glycine gels (Thermo Fisher, XP00100BOX) and subsequently transferred onto PVDF membranes (Bio-Rad, 162–0177) using the XCell II Blot Module (Thermo Fisher, EI9051). The antibodies used for Western blotting were rabbit anit-GFP (ab290, 1:7,500, abcam), guinea pig anti-D1 (1:1,000, same as above), HRP-conjugated goat anti-rabbit (1:10,000, abcam, ab97051) and HRP-conjugated goat anti-guinea pig (1:10,000, abcam, ab97155).

## Acknowledgements

We thank Cheng-Yu Lee, Georg Krohne, Tibor Torok, Bloomington Drosophila Stock Center, Kyoto Stock Center and Developmental Studies Hybridoma Bank for reagents and resources. We thank the Yamashita lab members for discussion and comments on the manuscript. This research was supported by the Howard Hughes Medical Institute (YY) and an American Heart Association postdoctoral fellowship (MJ). MJ and YY conceived the project, interpreted the data and wrote the manuscript. All authors contributed to conducting experiments and analyzing data.

## Additional information

### Competing interests

Yukiko M Yamashita: Reviewing editor, *eLife*. The other authors declare that no competing interests exist.

### Funding

| Funder | Author |
|---|---|
| Howard Hughes Medical Institute | Yukiko M Yamashita |
| American Heart Association | Madhav Jagannathan |

The funders had no role in study design, data collection and interpretation, or the decision to submit the work for publication.

### Author contributions

Madhav Jagannathan, Yukiko M Yamashita, Conceptualization, Formal analysis, Supervision, Funding acquisition, Investigation, Writing—original draft, Writing—review and editing; Ryan Cummings, Investigation, Writing—review and editing

### Author ORCIDs

Madhav Jagannathan http://orcid.org/0000-0003-3428-6812
Ryan Cummings http://orcid.org/0000-0003-0540-9174
Yukiko M Yamashita http://orcid.org/0000-0001-5541-0216

### Decision letter and Author response

Decision letter https://doi.org/10.7554/eLife.43938.037
Author response https://doi.org/10.7554/eLife.43938.038

## Additional files

### Supplementary files

• Transparent reporting form
DOI: https://doi.org/10.7554/eLife.43938.034

### Data availability

All data generated or analysed during this study are included in the manuscript and supporting files. Source data files have been provided for relevant figures.

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
