## [Decision Letter]

Thank you for submitting your article "The modular mechanism of chromocenter formation in *Drosophila*" for consideration by *eLife*. Your article has been reviewed by K VijayRaghavan as the Senior Editor, a Reviewing Editor, and two reviewers. The following individual involved in the review of your submission has agreed to reveal her identity: Susan A. Gerbi (Reviewer #2).

The comments are favorable, and some changes asked are speedily addressable. All best wishes for the season and the New Year,

Summary:

This is a fine paper worthy of publication in *eLife*. In a paper published earlier this year, the authors presented the novel idea that protein D1, which binds to the {AATAT}_n_ satellite DNA in *Drosophila*, serves to bundle chromosomes into chromocenters in nuclei and blebbing off of micronuclei occurs in the absence of D1. They demonstrated that the same effect is seen in mouse cells for the HMGA1 homologue of D1. Having established this conceptually new advance for the field, one could ask what more is new in the present manuscript. The answer to this question is positive, as significantly more information is presented in this manuscript.

While the concept remains unchanged, it is broadened by their new studies on the Prod protein that binds to the *Drosophila* {AATAACATAG}_n_ satellite DNA. They clearly show chromosome specificity in the binding: D1 binds to the {AATAT}_n_ that is located on chromosomes X, Y and 4, whereas Prod binds to {AATAACATAG}_n_ that is on chromosomes 2 and 3. In both cases, chromatin threads of protein binding to satellite DNA are seen that may link heterologous chromosomes together. Since *Drosophila melanogaster* has at least 17 different satellite DNAs, the question remains for future studies if there are other satellite-binding proteins and what their impact may be for chromocenter maintenance. Another unanswered question for future studies revolves around the rapid evolution of satellite DNA sequences that differ even between *Drosophila melanogaster* and its closely related species *Drosophila simulans*. Can the *D. melanogaster* Prod bind to satellite DNA in *D. simulans* (and vice versa), suggesting recognition of a feature beyond the simple sequence? Or, might species specificity of Prod binding to satellite DNA be at the heart of mechanisms that separate one species from another?

The authors use elegant *Drosophila* genetics approaches coupled with microscopy (including time lapse live cell imaging). The authors demonstrate tissue specificity of micronuclei formation with D1 mutation more severely affecting the germline whereas prod mutation more severely affects somatic cells. Moreover, genetic inhibition of apoptosis reveals DNA damage in prod mutant tissue. Their data suggest that D1 and Prod transiently interact for a reciprocal influence of DNA bundling. This leads to their model of a "modular network" for chromocenter formation. This explains the synthetic lethality of the *D1 prod* double mutant.

Essential revisions:

The data are mostly convincing, and the manuscript is well written and addresses an important topic: the function of pericentromeric satellite DNA. There are two concerns with the experiments showing interactions between D1 and Prod, which are crucial for the overall message.

First, the only direct evidence for interactions between D1 and Prod is their localization near each other (Figure 4A) and the fluctuating distance between them (Figure 4G). These results need to be carefully quantified and compared to distances between loci that are not expected to interact with each other. Otherwise the results are difficult to interpret.

Second, Figure 4H-M shows that the number of D1 foci increases when Prod is mutated and vice versa, suggesting that one of the satellite binding proteins support the clustering of the other. For this experiment to be convincing, FISH against the specific satellite DNAs should be included to rule out the possibility that the observed loss of clustering is due to satellite-binding proteins spreading to non-cognate sequences. Please see if this can be speedily done (which we think is feasible; but if not please make case why we can proceed without this experiment).

Coming from our editorial discussions, the following points also need to be addressed.

With regard to "transient interaction" between D1 and Prod, even with careful analysis (as suggested by the reviewer), the cytological evidence would only be suggestive at best. There was no co-IP seen, which the authors interpret as a transient interaction. The synthetic lethality of the D1/prod double mutant demonstrates a genetic interaction. Overall, the authors should soften the wording to conclude that there is suggestive evidence for interaction between D1 and Prod.

With regard to Figure 1B, the authors should explain in the legend both the arrows (denoting chromosome II(I) and the arrowheads (denoting chromosome II). Since the FISH probe was against the {AATAACATAG}_n_ satellite that resides on both chromosomes II and III, it is unclear how these two chromosomes could be distinguished in this particular experiment. It is only in Figure 3D where FISH probes are used against the chromosome 3 specific dodeca satellite or against the chromosome 2 specific {AACAC}_n_ satellite that chromosome 2 can be distinguished from chromosome 3. Finally, for consistency, the authors should use Arabic numbers to denote the chromosome identity (not Roman numerals as in Figure 1B).

---

## [Author Response]

Essential revisions:The data are mostly convincing, and the manuscript is well written and addresses an important topic: the function of pericentromeric satellite DNA. There are two concerns with the experiments showing interactions between D1 and Prod, which are crucial for the overall message.First, the only direct evidence for interactions between D1 and Prod is their localization near each other (Figure 4A) and the fluctuating distance between them (Figure 4G). These results need to be carefully quantified and compared to distances between loci that are not expected to interact with each other. Otherwise the results are difficult to interpret.

We understand reviewers’ concern. However, we feel that there is no appropriate ‘negative control’ (two loci that are expected to not interact) as suggested by reviewers. Any other satellite DNAs may be a part of ‘modular network’ that we propose in this study, and thus they likely cannot be used as a negative control. Other possibilities include the use of LacO repeats integrated into multiple loci and analyze their interaction. However, this will not serve as a fair control, because LacO repeats insertions (256 copies roughly spanning 10kb) are cytologically much smaller than the Mb-long tracts of AATAT or AATAACATG satellites. Therefore, it is expected that LacO repeats will exhibit less interaction than D1 and Prod, yet it does not prove the lack of interaction. Another possibility is the use of chromosome painting (such as oligo paint technique) of multiple loci. But this technique does not allow live observation, and thus we cannot assess dynamic interaction, and it is unlikely that we can utilize such data to infer their interaction (or lack thereof) to compare to that of D1 and Prod. Therefore, we decided to soften our statement throughout the manuscript so as not to mislead the readers.

Second, Figure 4H-M shows that the number of D1 foci increases when Prod is mutated and vice versa, suggesting that one of the satellite binding proteins support the clustering of the other. For this experiment to be convincing, FISH against the specific satellite DNAs should be included to rule out the possibility that the observed loss of clustering is due to satellite-binding proteins spreading to non-cognate sequences. Please see if this can be speedily done (which we think is feasible; but if not please make case why we can proceed without this experiment).

We appreciate this suggestion by the reviewers. This is indeed a very important point and we conducted this experiment in two parts, which is now added as Figure 4—figure supplement 3. The results presented in this figure demonstrate that D1 and Prod co-localize with AATAT/AATAACATAG satellite DNA, respectively, even when D1/Prod exhibit de-clustering in the absence of the other protein. Consistently, our quantification of AATAT satellite DNA foci in the absence of Prod and AATAACATAG satellite DNA foci in the absence of D1 in larval imaginal discs and lymph glands show that these satellite DNA foci exhibit similar levels of declustering to protein foci declustering as shown in Figure 4H-M.

Coming from our editorial discussions, the following points also need to be addressed.With regard to "transient interaction" between D1 and Prod, even with careful analysis (as suggested by the reviewer), the cytological evidence would only be suggestive at best. There was no co-IP seen, which the authors interpret as a transient interaction. The synthetic lethality of the D1/prod double mutant demonstrates a genetic interaction. Overall, the authors should soften the wording to conclude that there is suggestive evidence for interaction between D1 and Prod.

We agree with this comment and especially given the lack of additional analysis on this (as explained above), we have softened the claim regarding the Prod-D1 interaction throughout the manuscript.

With regard to Figure 1B, the authors should explain in the legend both the arrows (denoting chromosome II(I) and the arrowheads (denoting chromosome II). Since the FISH probe was against the {AATAACATAG}_n_ satellite that resides on both chromosomes II and III, it is unclear how these two chromosomes could be distinguished in this particular experiment. It is only in Figure 3D where FISH probes are used against the chromosome 3 specific dodeca satellite or against the chromosome 2 specific {AACAC}_n_ satellite that chromosome 2 can be distinguished from chromosome 3. Finally, for consistency, the authors should use Arabic numbers to denote the chromosome identity (not Roman numerals as in Figure 1B).

Thank you for pointing these out. For Figure 1B: Chromosome II and III were distinguished because chromosome III contains a minor amount of AATAT satellites. This was mentioned in the text but was not explained in the context of Figure 1B. Thus, we have added this explanation to the legend. We have changed all to Arabic numbers (figures and text).